# T Cell Mediated Conversion of a Non-Anti-La Reactive B Cell to an Autoreactive Anti-La B Cell by Somatic Hypermutation [note 1]

**DOI:** 10.3390/ijms22031198

**Published:** 2021-01-26

**Authors:** Michael P. Bachmann, Tabea Bartsch, Claudia C. Bippes, Dominik Bachmann, Edinson Puentes-Cala, Jennifer Bachmann, Holger Bartsch, Claudia Arndt, Stefanie Koristka, Liliana R. Loureiro, Alexandra Kegler, Markus Laube, Joanne K. Gross, Tim Gross, Biji T. Kurien, R. Hal Scofield, A. Darise Farris, Judith A. James, Marc Schmitz, Anja Feldmann

**Affiliations:** 1Department of Radioimmunology, Institute of Radiopharmaceutical Cancer Research, Helmholtz-Zentrum Dresden-Rossendorf (HZDR), 03128 Dresden, Germany; t.bartsch@hzdr.de (T.B.); epuentes@corrosion.uis.edu.co (E.P.-C.); c.Arndt@hzdr.de (C.A.); S.Koristka@hzdr.de (S.K.); l.loureiro@hzdr.de (L.R.L.); a.kegler@hzdr.de (A.K.); m.laube@hzdr.de (M.L.); a.feldmann@hzdr.de (A.F.); 2University Cancer Center (UCC), Tumor Immunology, University Hospital Carl Gustav Carus Dresden, Technical University Dresden, 01307 Dresden, Germany; domi_bachmann@hotmail.de (D.B.); bynnej@gmail.com (J.B.); 3National Center for Tumor Diseases (NCT), 01307 Dresden, Germany; 4German Cancer Research Center (DKFZ), 69120 Heidelberg, Germany; 5Institute of Immunology, Medical Faculty Carl Gustav Carus Dresden, Technical University Dresden, 01307 Dresden, Germany; claudia_bippes@gmx.de (C.C.B.); holger_bartsch@yahoo.de (H.B.); marc.schmitz@tu-dresden.de (M.S.); 6Corporación para la Investigación de la Corrosión (CIC), Piedecuesta, Santander 681011, Colombia; 7The Arthritis and Clinical Immunology Program, Oklahoma Medical Research Foundation and University of Oklahoma Health Sciences Center, Oklahoma City, OK 73104, USA; Jody-Gross@omrf.org (J.K.G.); Tim-Gross@omrf.org (T.G.); Biji-Kurien@omrf.org (B.T.K.); hal-scofield@omrf.org (R.H.S.); Darise-Farris@omrf.org (A.D.F.); Judith-James@omrf.org (J.A.J.)

**Keywords:** anti-La/SS-B antibodies, autoimmunity, La/SS-B autoantigen, systemic lupus erythematosus, primary Sjögren’s syndrome

## Abstract

Since the first description of nuclear autoantigens in the late 1960s and early 1970s, researchers, including ourselves, have found it difficult to establish monoclonal antibodies (mabs) against nuclear antigens, including the La/SS-B (Sjögrens’ syndrome associated antigen B) autoantigen. To date, only a few anti-La mabs have been derived by conventional hybridoma technology; however, those anti-La mabs were not bona fide autoantibodies as they recognize either human La specific, cryptic, or post-translationally modified epitopes which are not accessible on native mouse La protein. Herein, we present a series of novel murine anti-La mabs including truly autoreactive ones. These mabs were elicited from a human La transgenic animal through adoptive transfer of T cells from non-transgenic mice immunized with human La antigen. Detailed epitope and paratope analyses experimentally confirm the hypothesis that somatic hypermutations that occur during T cell dependent maturation can lead to autoreactivity to the nuclear La/SS-B autoantigen.

## 1. Introduction

Autoantibodies to nuclear antigens are frequently found in the sera of patients with systemic lupus erythematosus (SLE) and primary Sjögrens’ syndrome (pSS) [1]. As many as 30% of SLE patients produce IgG autoantibodies to the autoantigens Ro/SS-A (Sögrens’ syndrome associated antigen A) and La/SS-B (Sjögrens’ syndrome associated antigen) which seem to have been among the first detectable autoantibodies [2]. A higher prevalence of anti-La autoantibodies occurs in individuals suffering from pSS. Perhaps, the highest prevalence of these autoantibodies is found in mothers who have given birth to children with neonatal lupus (NL) [3].

The mechanisms leading to anti-La autoimmunity in pSS and SLE patients are still not completely understood. Up until now, many plausible mechanisms have been postulated for the development of autoimmunity including molecular mimicry, for example, with viral sequences [4,5,6,7,8,9], epitope spreading [10], impaired clearance of apoptotic cell material [11,12], and the generation of autoimmunity to cryptic or post-translationally modified epitopes (e.g., oxidized self) [13]. In addition, autoimmune responses have been postulated to be the result of somatic hypermutations occurring during maturation of originally non-autoreactive B cells [14]. These different mechanisms may not be mutually exclusive. 

For decades, we, like many other groups, have tried to trigger monoclonal antibody (mab) responses to nuclear autoantigens, including to the La/SS-B autoantigen, using conventional hybridoma technology [15,16,17,18,19,20,21,22,23,24,25,26,27]. For this purpose, usually, mice are repeatedly immunized with the respective antigen. Finally, splenocytes are isolated and fused with tumor cells derived from a myeloma cell line. The resulting hybridoma cells are screened and selected for secretion of the requested mab. More than fifty hybridoma fusions have been performed from mice that were immunized with either isolated human La, recombinant human La, or recombinantly expressed N- or C-terminal fragments of human La protein ([23,24,25,26,27], and Bachmann, unpublished). Other research groups have used, for example, purified bovine La protein for immunization [15,18]. Unfortunately, the heavy and light chain gene sequences of most of these hybridomas were not estimated or published. Epitope mapping data are also limited. The majority of anti-La mabs are directed conformational epitopes. Therefore, the epitopes recognized by these anti-La mabs are less well-defined epitopes. One such an example is the frequently used anti-La mab SW5 which recognizes a split epitope (aa 112–138 and aa 171–183) of human La protein, [15,16,22]). To the best of our knowledge, the only anti-La mabs which have been mapped and shown to recognize defined short continuous peptide epitopes are the 5B9 and 7B6 anti-La mabs [28]. The 5B9 anti-La mab recognizes the amino acid (aa) sequence KPLPEVTDEY (aa 95–104 of human La protein) which is conserved in human and mouse La protein. It is part of a random coiled region in the N-terminal domain of the La protein which connects the La motif with the RRM1 domain. The epitope is not accessible on native La protein as the 5B9 antibody fails to immunoprecipitate native La proteins (see also Results). The 7B6 anti-La mab recognizes the aa sequence EKEALKKIIEDQQESLNK (aa 311–328 of human La protein) which is part of a helical region (α3) in the C-terminal domain (RRM2) of the La protein [28]. In spite of the fact that the primary 7B6 epitope sequence is conserved between human and mouse La protein, the 7B6 anti-La mab usually does not recognize native murine La protein, most likely because the murine sequence contains an isoleucine (aa319) and a glutamic acid (aa320), which are replaced by threonine and aspartic acid, respectively. Moreover, the threonine in the mouse sequence is almost 100% phosphorylated, but the mab recognizes only the non-phosphorylated sequence. Furthermore, the 7B6 epitope is also cryptic and not accessible in native human La protein (see also Results). Both epitope sequences have been carefully characterized and confirmed in numerous studies by using them as peptide tags [28,29,30,31,32,33,34,35,36,37,38,39,40,41,42,43,44,45,46,47,48,49]. For example, the 7B6 sequence was introduced in the sequence of the extracellular domain of chimeric antigen receptor (CAR) genes and used for estimation of the transduction rate of CAR T cells and their enrichment by Fluorescence-Activated Cell Sorting (FACS ) [30]. Moreover, both modular CAR T cell platforms, UniCARs and RevCARs, are based on the 5B9 and 7B6 epitope/paratope combination [28,29,30,31,32,33,34,35,36,37,38,39,40,41,42,43,44,45,46,47,48,49]. 

In summary, all of the previously described anti-La mabs obtained by conventional hybridoma technology fail to cross-react with mouse La protein or only recognize denatured but not native mouse La protein. Consequently, none of these anti-La mabs are actually bona fide autoantibodies. 

In a previous report, Keech et al. were able to break tolerance in mice that were transgenic for the human La gene [50]. Tolerance was broken by adoptive transfer of T cells from non-transgenic mice which were immunized with human recombinant La protein, suggesting that anti-La tolerance is limited to the T cell compartment. However, so far, the autoimmune response to the La antigen has not been characterized at the molecular level.

In order to learn more about this T cell dependent anti-La B cell response, we established anti-La hybridomas from a human La transgenic mouse after adoptive transfer of T cells from non-transgenic litter mates that were immunized with recombinant human La protein. Unexpectedly, the hybridoma fusion resulted in a high number of anti-La hybridomas secreting antibodies (abs). Epitope and paratope analyses of the elicited anti-La hybridomas show that some of them originated from truly autoreactive B cells. Reverting the mutations in the primary amino acid sequence of an autoreactive anti-La ab back to its germline sequence result in an ab which no longer binds the La/SS-B autoantigen. Consequently, our data confirm experimentally the hypothesis that autoreactivity can be acquired during maturation due to T cell dependent somatic hypermutations.

## 2. Results

### 2.1. Hybridoma Fusion from a Human La Transgenic Mouse after Adoptive Transfer of T Cells from Non-Transgenic Mice Immunized with Human La

Over the past decades, our group and many others have tried to establish mabs to autoantigens by hyperimmunization of mice following conventional hybridoma technology (Figure 1A). Here, we established hybridomas from spleen cells of a human La transgenic mouse after adoptive transfer of T cells from non-La transgenic littermates which were immunized with human La protein (Figure 1B). The first unexpected result was the high number of ten anti-La positive hybridomas of which eight hybridomas could be successfully recloned to a monoclonal level. As summarized in Table 1, two anti-La mabs (13C5B and 16C) belonged to the IgM class and six were IgG type mabs (24BG7, 22A, 27E, 312B, 2F9, and 32A). The 27E anti-La mab is an IgG2b. All other IgG-type anti-La mabs belong to the IgG1 subclass. In addition to checking for reactivity against La protein, we also checked the cell culture supernatants for anti-Ro reactivity. One positive hybridoma in ELISA against the Ro60 antigen could be established (i.e., 312G). It belongs to the IgM subclass. The anti-Ro60 mab does not cross-react with La/SS-B.

### 2.2. The Hybridomas Originate from the La Transgenic Mouse

Our aim was to determine whether or not the established hybridomas actually originated from the human La transgenic mouse and were not accidently derived from adoptively co-transferred B cells of the non-transgenic mice which were hyperimmunized with human La protein. For that purpose, total cell extracts of the novel hybridomas were separated by SDS-PAGE and analyzed by immunoblotting. If the novel hybridomas were derived from the human La transgenic mouse, these hybridomas must contain both mouse as well as human La protein which have different mobilities, and thus can be separated by SDS-PAGE [26]. Otherwise, the hybridomas should only contain mouse La protein. As controls, we used extracts of the previously described and well characterized murine anti-La hybridomas SW5 [15,22], 7B6, and 5B9 which can only contain mouse La protein [27,28,29]. For detection of La proteins, we used the anti-La mab, 5B9, which recognizes both denatured human and mouse La protein (see also Introduction). As expected, the 5B9 anti-La mab reacted with mouse La protein in all analyzed hybridomas (Figure 2, mLa). In addition, it verified the presence of human La protein in all hybridomas that were generated from the La transgenic mouse (Figure 2, hLa), while human La protein was absent in the SW5, 7B6, and 5B9 hybridomas from non-transgenic mice. Thus, all the novel hybridomas indeed derived from spleen cells of the human La transgenic mouse and not from adoptively transferred anti-La B cells of the hyperimmunized non-transgenic mice. In line with these data, the 7B6 and SW5 anti-La hybridomas detected only one immunoreactive protein band according to the mobility of human La protein in the extracts of the anti-La hybridomas from the human La transgenic mouse but none in their own hybridoma cells (data not shown).

### 2.3. The Hybridomas Originate from Individual B Cells 

In order to learn whether the elicited hybridomas resulted from different B cells or represented a tree of the same B cell containing different numbers of hypermutations, we determined their heavy and light chain sequences. The nucleotide (Figure 3) and aa (Figure 4) sequences of the light (Figure 3A and Figure 4A) and heavy chains (Figure 3B and Figure 4B) of the anti-La mabs from the La transgenic mouse (+Tg) were both aligned and compared to the previously established anti-La hybridomas SW5, 7B6, and 5B9 obtained by conventional hyperimmunization (-Tg). The deviation of their respective nucleotide and aa sequences from their germline sequences were also estimated and included in the alignments highlighted in green in Figure 3 and Figure 4. In addition, the number of mutations at the nucleotide level and resulting aa replacements are summarized in Table 2 and Table 3 and graphically presented in Figure 5. According to the epitope mapping data (as described below), the anti-La mabs from the La transgenic mouse are arranged in two groups (Group 1, 24BG7 and 22A (shaded in grey in Figure 3 and Figure 4) and Group 2, 27E, 312B, 2F9, and 32A (shaded in blue in Figure 3 and Figure 4)). When looking at the VDJ recombination sites of the heavy chains (Figure 3B) encoding the respective CDR3 regions (Figure 4B), on the one hand, it becomes immediately evident that all the novel hybridomas are derived from individual B cells. On the other hand, the comparison also shows that at least some of the sequences are highly related. All the anti-La mabs from the La transgenic mouse (and interestingly also the non-La transgenic anti-La hybridoma 5B9) but one (i.e., 2F9) may have used a J558 V_h_ element for recombination (Table 1). Moreover, all the Group 2 anti-La mabs and also one of the two Group 1 anti-La mabs (22A) may have used the J_h_2 element for recombination. The other Group 1 anti-La mab (24BG7), similar to all the non-transgenic anti-La derived hybridomas, may have used the J_h_3 element for recombination. It is also interesting that all the IgM transgenic anti-La hybridomas and even the anti-Ro hybridoma used the J_h_4 element. A closer look at the light chain sequence shows that some of the anti-La hybridomas have even recombined the same V_l_ gene and the same joining element in the same manner. For example, the IgM anti-La mab ((13C5B) and the Group 2 anti-La hybridomas (27E and 32A) might have selected the same light chain element (V_l_ aa4) and the same joining fragment (JK1) (Figure 3A and Figure 4A, see also Table 1). Still, when looking at their heavy chains, they differ in their CDR3 regions and also in their CDR1 sequences. Bearing in mind that the light chain recombination follows the rearrangement of the heavy chain gene, these B cells must have independently selected the same V_l_- and J_l_-gene elements and recombined them in the same way. As evident from Table 1, the Group 1 anti-La mabs may have used the same V_l_-chain gene (kk4) but recombined it with different joining fragments (24BG7, JK2, 22A, and JK5). 

Taken together, the sequence analysis shows that all heavy and light chain sequences of the analyzed anti-La mabs, including those of the IgM type, show evidence of hypermutations (Table 2 and Table 3 and Figure 5). Interestingly, with a total number of three mutations, the IgG1 type anti-La mab (2F9) contains the lowest number of somatic mutations. Two of the three mutations at the nucleotide level occurred in its heavy chain and resulted in the replacement of two aa. Only one of these aa replacements occurred in a CDR region, namely the CDR2 region, while the other one occurred in the FWR1. The single mutation in the FWR3 of its light chain had no influence on its aa sequence. Thus, the aa sequence of the whole light chain, and also the CDR1 and CDR3 region of its heavy chain, have germ line configuration. On the one hand, even the obtained IgM type anti-La mabs contain more somatic mutations than the anti-La mab (2F9). On the other hand, with -5 nucleotides upstream and +13 nucleotides downstream of the D element, the contribution of n-nucleotides is highest in the anti-La mab (2F9). The Group 1 anti-La hybridomas (24BG7 and 22A) accumulated slightly more aa replacements than the Group 2 anti-La hybridomas. The Group 1 anti-La hybridoma (22A) contains the highest number of aa replacements of all anti-La hybridomas including the anti-La hybridomas (SW5, 7B6, and 5B9) from the three independent hybridoma fusions of the hyperimmunized, non-transgenic mice. With the previously mentioned exception of the Group 2 anti-La mab (2F9), all the other Group 2 anti-La mabs contain a comparable number of aa replacements, which is in the same range as the non-transgenic anti-La mab (5B9) which is less than the non-transgenic anti-La mabs (SW5 and 7B6). As previously mentioned, the IgM type anti-La mabs have undergone some but less somatic hypermutations (with the exception of the Group 2 anti-La mab (2F9) (Figure 5). The lowest number of somatic mutations and aa replacements of all analyzed mabs was found in the IgM type anti-Ro mab, suggesting that the T cell help was quite inefficient if at all occurred.

### 2.4. Reactivity of Anti-La Mabs to Denatured and Native Human and Mouse La Protein

#### 2.4.1. Analysis by SDS-PAGE and Immunoblotting Using La Proteins Recombinantly Expressed in *E. coli* and Total Extracts from Murine or Human Cell Lines

In order to further characterize the anti-La mabs obtained from the human La transgenic mouse, hybridoma supernatants were tested by SDS-PAGE and immunoblotting for reactivity against human La (Figure 6, rhLa) or mouse La (Figure 6, rmLa) both recombinantly expressed in *E. coli*, or eukaryotic human La (Figure 6, HeLa), or eukaryotic mouse La (Figure 6, 3T3). For an unknown reason, purified recombinantly expressed mouse La contains full length mouse La and a series of smaller La related fragments. All abs reacting to La antigen by ELISA also reacted with denatured human recombinant La protein by immunoblot. In agreement with previous data and also the data shown in Figure 2, the control anti-La mab (5B9) reacted with human and mouse La protein expressed in bacteria, as well as with eukaryotic human and mouse La protein. In agreement with previous data, the anti-La mabs (SW5 and 7B6) also recognized both bacterially expressed and eukaryotic human La protein but failed to react with eukaryotic mouse La protein. Interestingly, both anti-La mabs (7B6 and SW5) were able to react with recombinant mouse La protein, suggesting that the epitope recognized on mouse La protein was post-translationally modified in eukaryotic mouse La. This may also be the reason why infrequently the anti-La mabs (7B6 and SW5) showed a week reactivity with eukaryotic mouse La protein which may have occurred when the respective post-translational modification was partially removed (see also Figure 7, SW5). It may also explain why, originally, the anti-La mab SW5 had been described as an anti-La mab that cross-reacted with mouse La protein [15]. All IgG type anti-La mabs from the transgenic mouse reacted with both human La proteins expressed either in bacteria or eukaryotic cells. With the exception of the two anti-La mabs (22A and 32A), all the other anti-La mabs from the transgenic mouse including the anti-La mabs (24BG7, 2F9, 27E and 312B) also reacted with both bacterially expressed or eukaryotic mouse La protein, suggesting that, in contrast to the epitopes recognized by the anti-La mabs (SW5 and 7B6), the murine epitope of these anti-La mabs might not be post-translationally modified either in mouse or human La protein. In summary, these data show that the majority (four out of six) of the anti-La mabs from the transgenic mouse cross-react with both denatured mouse and human La protein. In addition, in both the Group 1 and the Group 2 anti-La hybridomas, we identified anti-La mabs that do or do not cross-react with denatured mouse La protein. 

#### 2.4.2. Coprecipitation Study of Native Human and Mouse La Protein from Total Cell Extracts of a Mouse 3T3 Cell Line Expressing the Human La Transgene

Next, we wanted to learn whether or not the anti-La mabs from the transgenic mouse also cross-reacted with native La proteins including mouse La protein. For this purpose, we prepared a total extract of a murine 3T3 cell line which was engineered to express the human La transgene. In agreement with the data described above, the mouse and human La protein present in the total extract of this human La transgenic mouse 3T3 cell line could be separated by SDS-PAGE and could be detected by immunoblotting with all the anti-La mabs shown to cross-react with both mouse and human La protein, including the anti-La mabs (5B9, 24BG7, 2F9, 27E, and 312B), while the other anti-La mabs including 7B6, SW5, 22A, and 32A did only react with the human La protein (data not shown). When the same total extract (Figure 7, TE) was used for coprecipitation of native La proteins, the Group 2 anti-La mab (312B) strongly coprecipitated both native mouse and human La protein. Although to a weaker extend, also the anti-La mab (27E) was able to coprecipitate both native human and mouse La protein. In contrast, the other two, Group 2 anti-La mabs (2F9 and 32A) failed to coprecipitate native human and mouse La protein. Similar to the anti-La mab (SW5), the Group 1 anti-La mab (22A) strongly coprecipitated human La protein and faintly mouse La protein. The Group 1 anti-La mab (24BG7) also coprecipitates native La proteins, although to a much lower extent than, for example, the anti-La mab (312B). Taken together, the two anti-La mabs (27E and 312B) are able to coprecipitate native human and mouse La protein, and thus represent, per definition, anti-La monoclonal autoantibodies. As the mouse was transgenic to human La protein, the 22A anti-La mab also represents an autoantibody, although it recognizes predominantly human La protein. 

#### 2.4.3. Immunofluorescence Analysis Using Human and Murine Cell Lines

For epifluorescence analysis, human HeLa (Figure 8) and murine 3T3 cell lines (Figure 9) were fixed either with paraformaldehyde (PFA) or methanol (MeOH/EGTA) and stained with hybridoma supernatants followed by a secondary anti-mouse-IgG ab conjugated with Alexa Fluor^®^568. Nuclei were counterstained with DAPI. As expected, the control anti-La mabs (SW5 and 5B9) gave an almost exclusive nuclear staining pattern on human HeLa cells. The fixation procedure did not remarkably influence their staining pattern. In contrast, the control anti-La mab 7B6 failed to stain human HeLa cells. From independent studies, we know that more stringent fixation conditions are necessary for accessibility of the 7B6 epitope (to be published). However, for reasons of comparability, the fixation and staining procedures were performed in parallel and in the same way for all abs. The Group 2 anti-La mabs (27E and 312B) gave a nuclear staining pattern comparable to the control anti-La mabs (SW5 and 5B9). The staining was not dependent on the fixation technique. The Group 1 anti-La mabs (24BG7 and 22A) also gave a comparable nuclear staining pattern on human HeLa cells. However, the staining failed in the case where the cells were fixed with MeOH/EGTA. In contrast to the two autoreactive Group 2 anti-La mabs (27E and 312B), the other two Group 2 anti-La mabs (2F9 and 32A) gave only a faint staining of human HeLa cells. As shown in Figure 9, the anti-human specific control anti-La mab (SW5) (and also 7B6) did not stain murine 3T3 cells, while the anti-mouse La cross-reactive control mab (5B9) gave the typical nuclear staining pattern. However, the staining was dependent on the fixation. Only 3T3 cells were stained which were fixed with MeOH/EGTA. Vice versa, the anti-mouse reactive Group 1 anti-La mab (24BG7) stained mouse 3T3 cells that were fixed with PFA but not cells fixed with MeOH/EGTA. The two autoreactive Group 2 anti-La mabs (27E and 312B) also stained the mouse 3T3 cells but the cells had to be fixed with PFA. The other two Group 2 anti-La mabs (2F9 and 32A) failed to stain irrespective of the fixation procedure. 

In addition to the IgG type anti-La mabs, we also used the obtained IgM type mabs for epifluorescence microscopy (Figure 10). Irrespective of their reactivity in ELISA, all IgM type mabs, including the anti-Ro mab, gave an atypical non-specific but strong cytoplasmic staining. The staining was not dependent on the fixation procedure and all the mabs stained both human HeLa and murine 3T3 cells equally well. Consequently, all these IgM type mabs are polyreactive and clearly different from the IgG type mabs.

### 2.5. Epitope Mapping 

In analogy to previous studies, epitope mapping was performed using N- and C-terminally truncated deletion mutants of the La protein. The results of the epitope mapping are summarized in Figure 11, Figure 12 and Figure 13. For mapping of the IgG type anti-La mabs, a series of green fluorescent protein (GFP) fusion constructs of La protein were cloned representing N- or C-terminally truncated deletion mutants, as listed in Materials and Methods. After transfection into human HeLa cells, total extracts were prepared and used for SDS-PAGE and immunoblotting. GFP expression levels and endogenous La protein were used as internal standards. In a first round, all anti-La mabs were screened on all deletion mutants. Thereby, we identified the most informative La deletion mutants. In a second round, the anti-La mabs were tested against these sets of deletion mutants (Figure 11 and Figure 12). In order to confirm the identified epitope region, the respective deletion mutant was constructed containing all N- and C-terminal aa required for immunoreactivity (Figure 11 and Figure 12, epitope region). According to epitope mapping, the anti-La mabs were sorted into two groups. As shown in Figure 11, both Group 1 anti-La mabs react with the final La deletion mutant La_107–200_. As shown in Figure 12, all Group 2 anti-La mabs recognize the final La deletion mutant La_10–100_. Any further deletion from either the N- or the C-terminal site results in lack of reactivity. Consequently, the La epitope(s) recognized by the Group 1 anti-La mabs (22A and 24BG7) are part of the RRM1 domain, while the La epitope(s) recognized by the Group 2 anti-La mabs (2F9, 32A, 27E, and 312B) are part of the La motif.

We also tried to map the IgM type mabs. However, they showed no specific immunoreactivity on extracts of transfected or non-transfected human or mouse cell extracts (data not shown). They also failed to react with purified recombinant proteins expressed in *E. coli*. Only the anti-La mab 16C reacted after SDS-PAGE and immunoblotting. However, for reactivity ten times the usual amount of La protein or deletion mutants had to be applied (Figure 13). According to these data, the epitope of the 16C may be located in the C-terminal domain of La protein. As the shortest fragment with immunoreactivity, we identified La_278–408_.

### 2.6. Lack of Anti-La Reactivity of the 312B Germline Sequence 

According to the coprecipitation experiment, the 312B anti-La mab effectively recognizes both native human and mouse La protein (see also Figure 7). In order to learn whether or not the ab acquired autoreactivity during somatic hyper mutation, we restored the germline sequence of the heavy and light chain sequences (Figure 14A,B). For this purpose, comparable recombinant derivatives based on the germline and the mature 312B sequence were cloned and analyzed by SDS PAGE and immunoblotting. In a first step, the respective scFvs were constructed and fused on the same Ig4 heavy chain backbone. The resulting ab is schematically shown in Figure 14C. The recombinant abs were eukaryotically expressed and purified (Figure 14D). While the ab based on the mature 312B sequence nicely recognized recombinant human La protein, the respective germline derivative failed to do so (Figure 14E). The lack of affinity of the 312B germline ab to La protein was confirmed by surface plasmon resonance analysis (data not shown).

## 3. Discussion

The presence of autoantibodies against nuclear antigens such as the La/SS-B autoantigen is a well-known and accepted hallmark in sera of patients with systemic autoimmune diseases such as SLE and pSS. Over the past decades, we like many other groups have tried to establish mabs, including against La/SS-B, using conventional hybridoma technology. In spite of many attempts, only a few mabs have become available and most, if not all, are not truly autoreactive but directed to species specific, cryptic, or post-translationally modified epitopes. Many of the obtained mabs show cross-reactivities. Even for the well-established anti-La mab (SW5), Fouraux et al. described a cross-reactivity with early endosomal protein 2 [51]. Many ideas for breaking tolerance to nuclear antigens have been put forward. It has been postulated that polyreactive T cells provide help to polyreactive B cells leading to somatic hypermutations which convert polyreactive IgM type abs to highly specific autoreactive IgG abs [52,53]. Even abs that are non-reactive in germline configuration have been discussed to be converted by hypermutations to autoreactive abs [14]. Unfortunately, anti-La mabs obtained by conventional hybridoma technology are rarely, if at all, bona fide autoantibodies. Moreover, the number of hybridomas elicited from a single experimental mouse, so far, has been too low to follow the fate of a polyreactive B cell on its way to a highly specific B cell clone caused by somatic hypermutations. In 2001, Keech et al. described a break of tolerance in mice transgenic for human La protein by adoptive transfer of T cells from non-La transgenic mice that were immunized with recombinant human La protein [50]. The break of tolerance was dependent on the adoptively transferred anti-La reactive T cells from non-transgenic mice immunized either with recombinant human La protein in the presence of complete Freund’s adjuvant (CFA) or heterologous neopitopes present in late apoptotic cells [54]. In order to further characterize this anti-La immune response, we decided to perform a hybridoma fusion from a La transgenic mouse that had obtained adoptively transferred T cells from non-transgenic mice which were immunized with recombinantly expressed human La protein. The first unexpected result was the relatively high number of anti-La hybridomas. As all the hybridomas contained the human La transgene, the anti-La hybridomas had to be derived from B cells of the La transgenic mouse. All the IgM type mabs gave an atypical cytoplasmic staining, while all the IgG type mabs gave the typically expected nuclear staining pattern (Table 4). Originally, we speculated that the IgM hybridomas may represent polyreactive precursors of the IgG B cell hybridomas which may have been converted during T cell dependent maturation into several anti-La IgG hybridomas by somatic hypermutation. As also summarized in Table 4, some of the anti-La mabs cross-reacted with native mouse La protein, while others recognized only human La protein, therefore, we hoped to learn how autoreactive anti-La B cells can develop from B cells reactive to foreign or cryptic La epitopes. However, our detailed sequence analysis of all the obtained anti-La mabs provided an unexpectedly different picture. All the anti-La hybridomas may go back to independent VDJ and VJ recombinations of their individual heavy and light chain genes and their combinations. This becomes obvious when looking at the CDR3 regions of the anti-La mabs and is supported by the detailed sequence and epitope analysis. Although the hybridomas may go back to individual clones, the sequence supports the idea that both the truly autoreactive anti-La B cells and the anti-La B cells recognizing cryptic epitopes originate from a common pool of precursor B cells [55].

Epitope mapping was performed using N- and C-terminally truncated La deletion mutants. In order to identify post-translationally modified epitopes, the deletion mutants were expressed both in bacteria and in eukaryotic cells. Using deletion mutants, two types of epitopes were differentiated and further characterized, namely continuous linear epitopes and conformational epitopes. The previously described anti-La mabs (7B6 and 5B9) are prototypes of mabs being directed to short continuous peptide epitopes, while the anti-La (SW5) is directed against a conformational epitope. Continuous epitopes can be refined to short peptide sequences which can be further characterized using, for example, synthetic peptides or epitope-fusion proteins containing His-tagged or GFP-epitope fusion constructs. Finally, the epitope sequence can be confirmed by using the identified aa sequence as a peptide tag. In the case of a conformational epitope, however, only the N- and C-terminal regions required for immunoreactivity can be identified using deletion mutants. For example, Veldhoven et al. located the conformational epitope region recognized by the anti-La mab (SW5) (and the related anti-La mabs (SW1 to 3)) to aa 112–183 of the La protein [16]. Using our N- or C-terminally truncated GFP-La deletion mutants, we could confirm the previously identified epitopes recognized by the anti-La mabs (SW5, 7B6, and 5B9) (data not shown), highlighting that the applied epitope mapping procedure works for both continuous and conformational epitopes. Noteworthy to mention, the anti-La mab (SW5) recognizes mouse La protein when expressed in *E. coli* but not when expressed in eukaryotes, indicating that the murine epitope is post-translationally modified in eukaryotic cells. 

Applying the same procedure, we were not able to clearly map the epitopes recognized by the polyreactive IgM type anti-La mabs. In contrast, for all the IgG type anti-La mabs, we identified conformational epitopes. The epitope region recognized by the Group 1 anti-La mabs was estimated with La_107–200_. The epitope region recognized by the Group 2 anti-La mabs was estimated with La_10–100_. Consequently, the Group 1 anti-La mabs are directed against the RRM1 domain of La protein, while the Group 2 anti-La mabs recognize the La motif domain. Thus, the Group 1 anti-La mabs recognize a similar epitope fragment as the previously described anti-La mab (SW5). It is also interesting to mention that the C-terminal portion required for reactivity of the Group 2 anti-La mabs represents the continuous epitope sequence recognized by the anti-La mab (5B9). When looking at the predicted 3D structure of the La motif and the RRM1 domain, it becomes evident that the La motif domain and the RRM1 domain both are folded in the way that the identified respective N- and C-terminal portions of the conformational epitopes come into close vicinity and could represent portions of a split epitope (Figure 11 and Figure 12). However, bearing in mind the huge differences with respect to their different reactivities to denatured or native La proteins or to human La protein alone or to both human and mouse La protein, it may also be possible that the identified N- and C-terminal portions are required to stabilize the three-dimensional structure of both domains. Deletion of any additional N- or C-terminal aa may simply lead to an unfolding of these domains and, as a consequence, to an alteration of the actual internally located conformational epitopes. Then, all anti-La mabs could easily recognize individual different conformational epitopes which are normally stabilized by the identified N- and C-terminal regions of the respective immunoreactive La fragment. For example, the La RRM1 fragment contains, at both the N- and C-terminus, a helical domain which may help to stabilize this La fragment. Deletion of an N- or C-terminal aa could destabilize the respective helix and result in an unfolding of the whole RRM1 region, including the internally located conformational epitopes. A deletion at the N-terminal helix of the La motif could cause a similar effect.

Obviously, the polyreactive IgM mabs stained non-specifically human and even mouse cells. Their cytoplasmic staining is similar, in part, to the staining of the IgM type anti-Ro mab previously described by us in 1986 [23]. Unfortunately, none of the IgM type mabs showed any specific immunoreactivity on extracts of either transfected or non-transfected human or mouse cells (data not shown). With the exception of the 16C anti-La mab, all of the IgM type mabs also failed to react with purified recombinant proteins expressed in *E. coli*. Only the 16C IgM mab reacted after SDS-PAGE and immunoblotting. However, for reactivity, ten times the usual amount of La protein or deletion mutants had to be applied to the gel (Figure 13). According to these data, the epitope of the 16C may be located in the C-terminal domain of La protein. We identified La_278–408_ as the shortest fragment with immunoreactivity. Although these epitope mapping data are limited, they show that the reactivity of at least the 16C IgM type mab might be unrelated to the epitopes recognized by the IgG type anti-La mabs. It still remains open whether or not these polyreactive IgM mabs can mature into highly specific autoreactive anti-La mabs of the IgG type. 

Our finding that the germline counter part of the 312B anti-La mab fails to recognize the La antigen, tells us that even non-anti-La reactive B cells can acquire anti-La autoreactivity as a consequence of somatic hypermutation. At least in the case of the 312B anti-La mab, the inserted n-nucleotides might not play a major role, as we did not change them when we reverted the 312B heavy and light chain sequences into the respective germline sequence.

Which of the mutations or combination of mutations finally lead to the autoreactivity is still not confirmed. It is tempting to speculate that the D to Y mutation in the CDR3 region of the heavy chain of 312B may play a prominent role, however, future studies in which the germline sequence is converted step-by-step into the 312B sequence will hopefully help to further clarify the contribution of each of the somatic mutation on the way from non-reactivity to autoreactivity. 

## 4. Materials and Methods

### 4.1. Recombinant Human La Protein Expression and Characterization

For immunization of non-La transgenic mice, recombinant human La protein was expressed in *E. coli* and purified, as described previously [50]. For ELISA screening of hybridoma supernatants, SDS-PAGE, and immunoblotting, human full length La protein aa 1 to 408 was recombinantly expressed in *E. coli* using a previously described La pET-3d construct containing the C-terminally His-tagged full length open reading frame of the human La protein (hLa, La_1–408_) [56]. For analysis of cross-reactivity with murine recombinant La protein, the mouse La (mLa) open reading frame was cloned into the expression vector pET28b. For first epitope mapping analysis and for mapping of the IgM type anti-La mab 16C epitope, deletion mutants representing La_1–126_, La_1–192_, La_1–245_, La_1–371_, La_159–408_, La_194–408_, La_239–408_, La_278–408_, La_318–408_, La_369–397_, and La_369–408_ were cloned into pET28a and expressed in *E. coli.* Isolation of His-tagged recombinant La proteins and deletion mutants were performed using nickel-NTA affinity chromatography [26,56]. To identify potential eukaryotic post-translational modifications, GFP-La fusion constructs were cloned into the vector pEGFP-C2 encoding either full length human La protein or the following deletion mutants: La_1–94_, La_1–100_, La_5–100_, La_10–100_, La_15–100_, La_20–100_, La_86–100_, La_1–104_, La_90–104_, La_94–104_, La_1–112_, La_82–112_, La_1–126_, La_1–192_, La_1–245_, La_1–310_, La_1–344_, La_107–200_, La_120–245_, La_194–408_, La_278–408_, La_295–408_, La_318–408_, La_346–408_, La_369–397_, La_369–408_, La_376–408._ The respective La reading frames were amplified by PCR. The GFP-La fusion constructs were transfected into human HeLa cells. Transfected cells were analyzed by epifluorescence microscopy. Prior to epifluorescence microscopy, cells were fixed with either paraformaldehyde (PFA) or methanol (MeOH). Fixed cells were stained with hybridoma cell culture supernatants and analyzed, as described previously [57]. Alternatively, total cell extracts were prepared and used for SDS-PAGE and immunoblotting [58]. For coprecipitation analysis, we used anti-mouse IgG TrueBlot^TM^ beads. Incubation with mabs and isolation of coprecipitated immune complexes were performed, according to the manufacturer’s instruction (eBioscience, San Diego, CA, USA). Coprecipitated immune complexes were analyzed by SDS-PAGE and immunoblotting. Anti-La and anti-Ro ELISA were performed, as described previously [59].

### 4.2. Mice

All mice were housed in specific pathogen free conditions, and the OMRF IACUC approved all mouse-related studies (Approval number B0042, 14 May 2001). The A/J mice transgenic for the wild type human La gene, including its natural promoter, have been described [50], and we used them as heterozygotes at the ≥F12 backcross generation. Mice were genotyped, as previously described [50]. 

### 4.3. T Cell Isolation, Adoptive Transfer, and Hybridoma Fusion

Immunization of non-transgenic litter mates with recombinant human La protein, isolation of T cells, and adoptive transfer of T cells was performed, as described previously [50]. In summary, donor mice were immunized with 100 micrograms 6× his-human recombinant La protein in complete Freund’s adjuvant (Difco, Lawrence, KS, USA) delivered in a divided dose subcutaneously in one hind footpad and the base of the tail. Seven days later, T cells were enriched from draining inguinal and popliteal lymph nodes by nylon wool chromatography from four animals. Residual B cells and other antigen presenting cells were depleted using a 1:1 mixture of Dynabeads (Dynal, Great Neck, NY, USA) coated with anti-B220 (Dynal 114.01) and anti-rat IgG Dynabeads coated with anti-mouse I-A/I-E antibody (Clone 10.2.16). Such isolated cells should represent a mixture of both CD4+ and CD8+ T cells but were not further characterized. The recipient La transgenic mouse received 5 million of the isolated T cells intraperitoneally. As the control, T cells were isolated from similarly treated animals with the only difference that the immunization mixture lacked recombinant human La protein. Isolated T cells were also adoptively transferred in another La transgenic mouse. Hybridomas were prepared from both animals. All the herein described anti-La hybridomas were obtained from the La transgenic recipient mouse which received the T cells isolated from non-transgenic mice after immunization with recombinant human La protein. In contrast, we could not isolate a single anti-La hybridoma from the control mouse, indicating that the anti-La reactive B cells were attracted to the spleen by the adoptively transferred anti-La reactive T cells present in the T cell preparation isolated from the non-La transgenic mice after immunization with human recombinant La protein.

### 4.4. Hybridoma Fusion

Hybridoma fusion, screening for La- and Ro-positive clones, and subcloning was performed, as described previously [23,24,25,57,59].

### 4.5. Sequence Analysis

cDNA-synthesis was performed using the Advantage™ RT-for-PCR Kit (TaKaRa Bio Europe, Göteborg, Sweden). For PCR we used the Advantage™ HF2 PCR Kit (TaKaRa Bio Europe). For amplification of the light chain sequences, we used the combination of the forward primer LC*for* (TTTTTGAATTCT **GAYATTGTGMTSACMCARWCTMCA)** with the reverse primer LCκ*rev* (TTTTTGGGCCCGGATACAGTTGGTGCAGCATC). For amplification of the heavy chain sequences, we used as common forward primer HC*for* (TTTTTGGATCC**SARGTNMAGCTGSAGSAGTCWGG**). In dependence on the Ig class or subclass HC*for* was combined with one of the following reverse primers either IgG1*rev* (GGAAGATCTATAGACAGATGGGGGTGTCGTTTTGGC), or IgG2A*rev* (GGAAGATCTCTTGACCAGGCATCCTAGAGTCA), or IgG2B*rev* (GGAAGATCTAGGGGCCAGTGGATAGACTGA) or IgM*rev* (ATTGGGACTAGTTTCTGCGACAGCTGGATT). Estimated variable heavy (V_h_) and light chain (V_l_) sequences were compared against the germ line sequences present in the NCBI data library using IgBlast sequence analysis tool. In order to calculate the number of somatic hypermutations, the respective V_l_ element with the highest sequence homology was selected. In a few cases (the heavy chains of 7B6, 5B9, 24BG7, and 27E) it became obvious that the second or even third best fitting sequence represented the originally used V_h_ element, and therefore this sequence was used for the calculations. Framework regions (FWR) and complementarity determining regions (CDR) were annotated according to Kabat database [60]. Sequence alignments were prepared using the AlignX-module of program Vector NTI.

For 3D modelling of the La-motif and RRM1 of La protein we used the program MOLMOL and the published NMR data (PDB: 1S7A La-motif, 1S9A RRM1, 1OWX RRM2) [61,62].

### 4.6. Construction, Expression, and Purification of Recombinant Antibodies

The recombinant 312B Ig4 derivatives were constructed as follows: The variable light (V_L_) and heavy (V_H_) chains of the monoclonal antibody (mAb) 312B were arranged in a V_H_-V_L_ orientation and connected by a (Gly_4_Ser)_3_ linker. The variable regions were further linked downstream to the hinge and Fc region (C_H_2-C_H_3) derived from a human IgG4 antibody [44,45,46]. An Igκ leader peptide (LP) sequence was fused to the N-terminus to promote secretion of the abs into the cell culture supernatant. The final sequences were cloned into the vector p6NST50 and used for transduction to generate stable ab-producing 3T3 cell lines. Recombinant abs were purified from cell culture supernatant via protein A affinity chromatography, according to manufacturer’s instructions (Protein A HP Spin Trap, Sigma-Aldrich, Taufkirchen, Germany), followed by overnight dialysis in PBS (Biochrom, Berlin, Germany). Finally, SDS–PAGE and immunoblotting were used to assess potential contaminants and to determine protein concentration, as previously described [44,45,46].

## Figures and Tables

**Figure 1 ijms-22-01198-f001:**
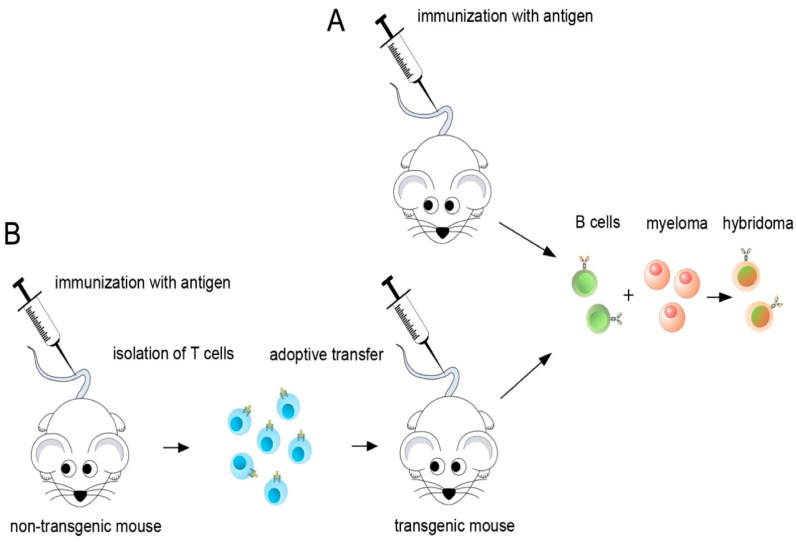
Development of monoclonal antibodies (mabs). (**A**) Conventional hybridoma technology starts with repeated immunizations of mice with the antigen. Isolated spleen cells are fused with myeloma cells resulting in hybridomas. Cell culture supernatants of the hybridomas are screened for the presence of the requested mab. Cells secreting the ab of interest are subcloned to a single cell level; (**B**) As in (**A**), we also started with immunization of (non-transgenic) mice. However, T cells were isolated from spleens of the immunized animals and adoptively transferred into human La transgenic animals, as described previously [50]. The serum of the mouse was tested for anti-La mabs. After seroconversion, the spleen was isolated and the hybridoma fusion was performed, as schematically summarized in (**A**).

**Figure 2 ijms-22-01198-f002:**
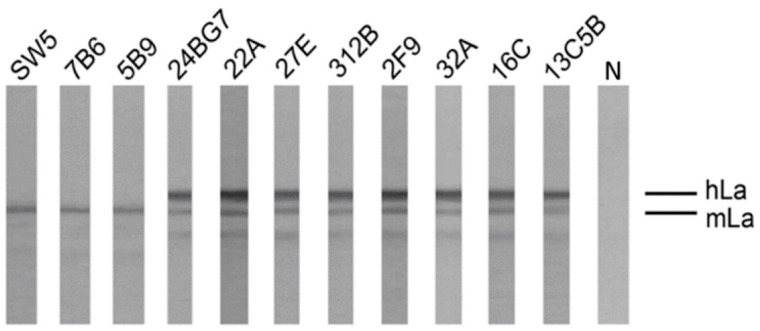
Origin of novel hybridomas. In order to rule out that the anti-La hybridomas, described here, are derived from adoptively co-transferred B cells of the immunized non-transgenic mice, the hybridomas were analyzed for the presence of human La protein using SDS-PAGE and immunoblotting. For this purpose, total extracts of the anti-La hybridoma cells including the previously established and well characterized conventional anti-La hybridomas (SW5, 7B6, and 5B9) and in addition the isolated anti-Ro hybridoma (312G) were prepared and blotted against the anti-La mab (5B9) which recognizes a cryptic epitope present in both human (hLa) and murine La protein (mLa). The secondary ab served as negative control (N).

**Figure 3 ijms-22-01198-f003:**
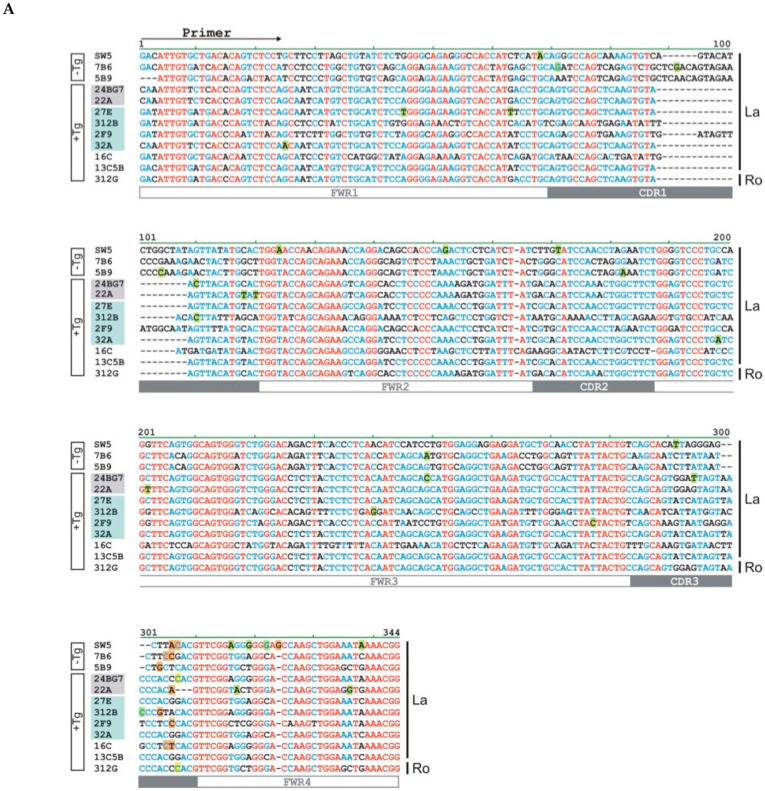
Nucleotide sequence alignment. The nucleotide sequences of (**A**) light and (**B**) heavy chains of the mabs were estimated and aligned. The -Tg sequences of the previously established non-transgenic hybridomas (SW5, 7B6, and 5B9). The +Tg sequences of the mabs established from the human La transgenic mouse including the Group 1 anti-La hybridomas (highlighted in grey), Group 2 anti-La hybridomas (highlighted in blue), as well as the IgM type mabs positive in ELISA against recombinant La protein (16C and 13C5B) or Ro60 (312G). FWR, framework region; CDR, complementarity determining region; Primer, forward primer used for PCR. Sequence differences in the primer region were not included for evaluation of somatic hypermutations as these differences may be derived from the used degenerated primers. Somatic mutations leading to aa replacements are highlighted in green.

**Figure 4 ijms-22-01198-f004:**
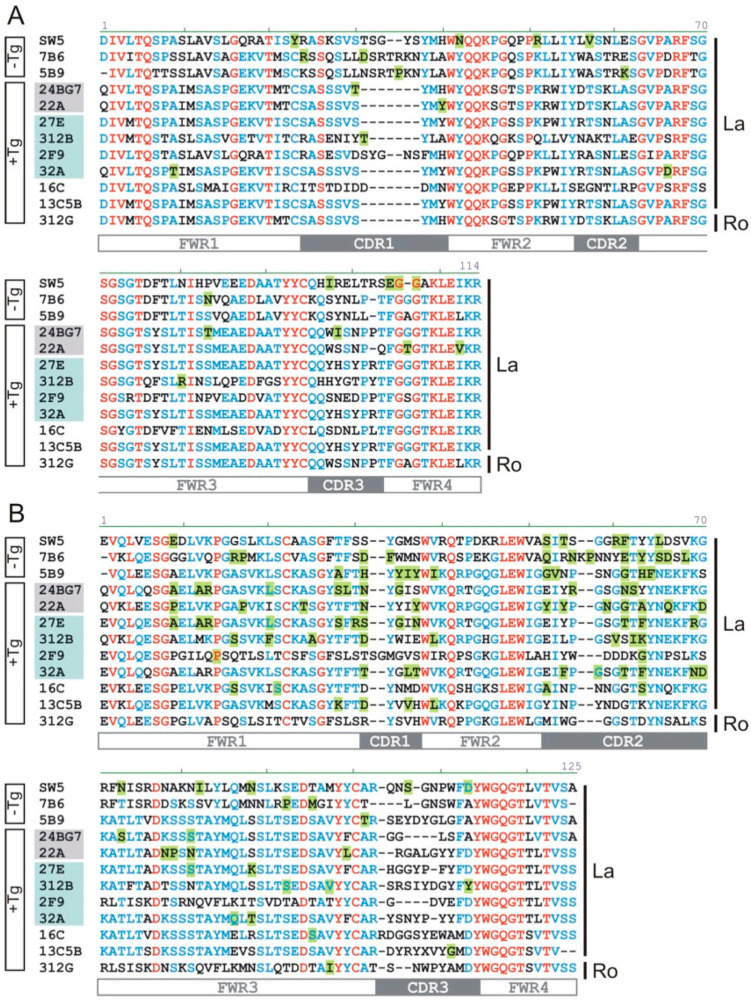
Amino acid sequence alignment. The aa sequences of (**A**) light and (**B**) heavy chains of the mabs were estimated and aligned. The -Tg sequences of the previously established non-transgenic hybridomas (SW5, 7B6, and 5B9). The +Tg sequences of the mabs established from the human La transgenic mouse including the Group 1 anti-La hybridomas (highlighted in grey), Group 2 anti-La hybridomas (highlighted in blue), as well as the IgM type mabs positive in ELISA against recombinant La protein (16C and 13C5B) or Ro60 (312G). FWR, framework region; CDR, complementarity determining region. Sequence differences detected in the primer region were not included for evaluation of somatic hypermutations as these differences may be derived from the used degenerated primers. aa replacements are highlighted in green.

**Figure 5 ijms-22-01198-f005:**
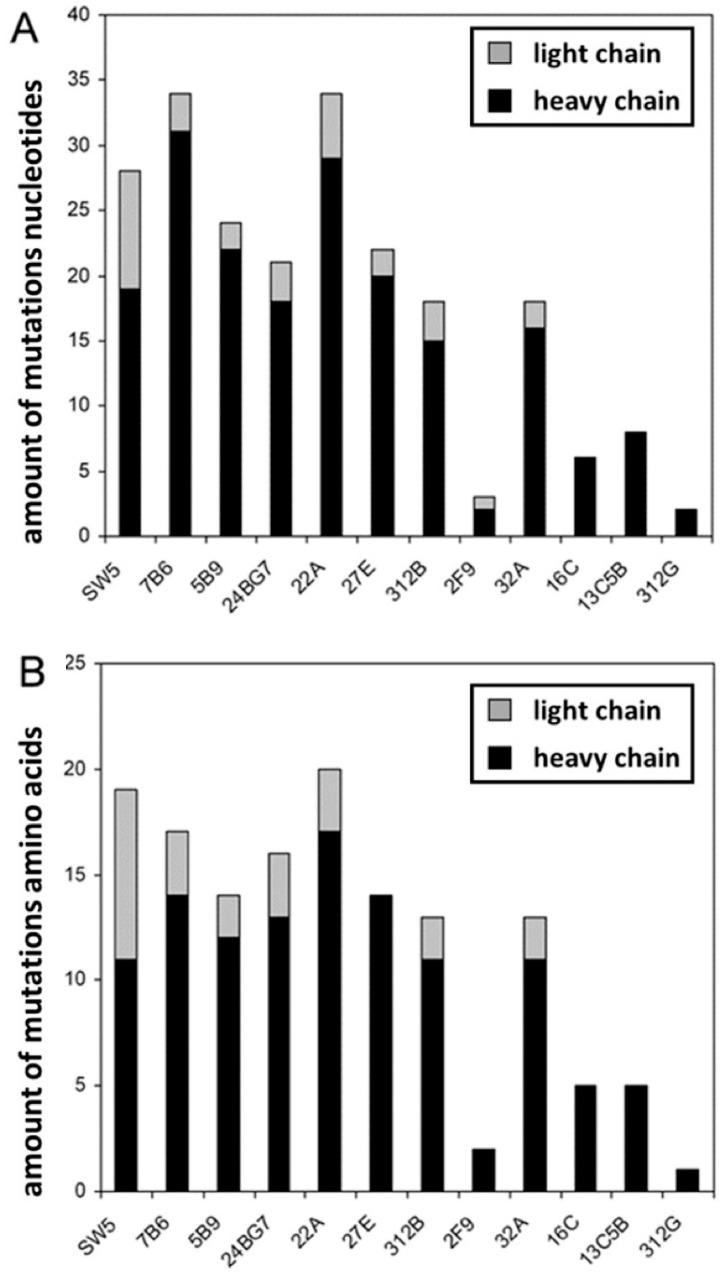
Estimation of mutations in the mabs. The total number of hypermutations present in the sequences of the light (grey) and heavy (black) chains were calculated for the respective (**A**) nucleotide or (**B**) aa sequence. Sequence differences in the region of the degenerated primers were not included.

**Figure 6 ijms-22-01198-f006:**
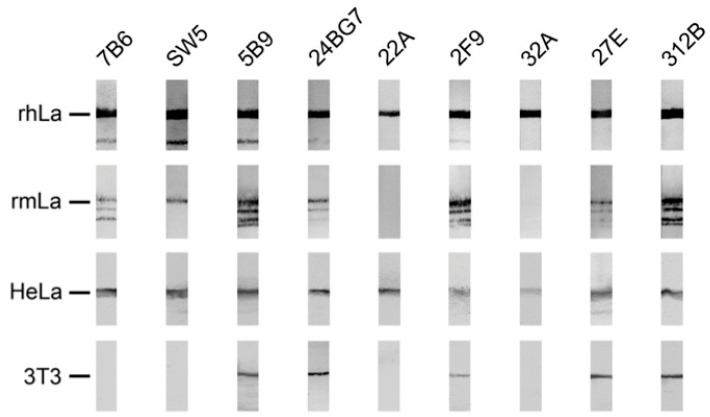
Reactivity of mabs to human and mouse La protein analyzed by SDS-PAGE and immunoblotting. Human (rhLa) or mouse (rmLa) La protein recombinantly expressed in *E. coli* or total extracts from eukaryotic human (HeLa) or mouse (3T3) cells were analyzed by SDS-PAGE and immunoblotting.

**Figure 7 ijms-22-01198-f007:**
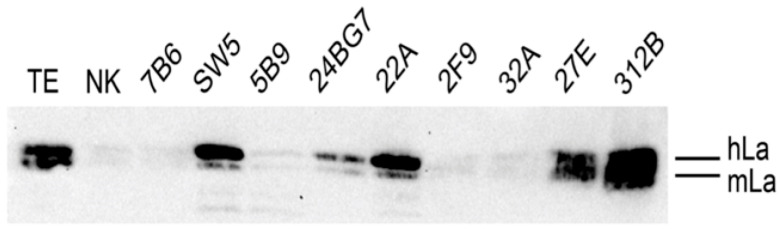
Immunoprecipitation of native human and mouse La protein. Total extract (TE, positive control) prepared from mouse 3T3 cells overexpressing the human La transgene was incubated with the respective anti-La mab. The formed immune complexes were coprecipitated using anti-mouse IgG TrueBlot^TM^ beads. Immune complexes were analyzed by SDS-PAGE and immunoblotting. Coprecipitated La proteins were detected using the 5B9 anti-La mab which recognizes equally well human and mouse La protein. N, negative control. As negative control TE was coprecipitated with anti-mouse IgG TrueBlot^TM^ beads in the absence of any anti-La mab.

**Figure 8 ijms-22-01198-f008:**
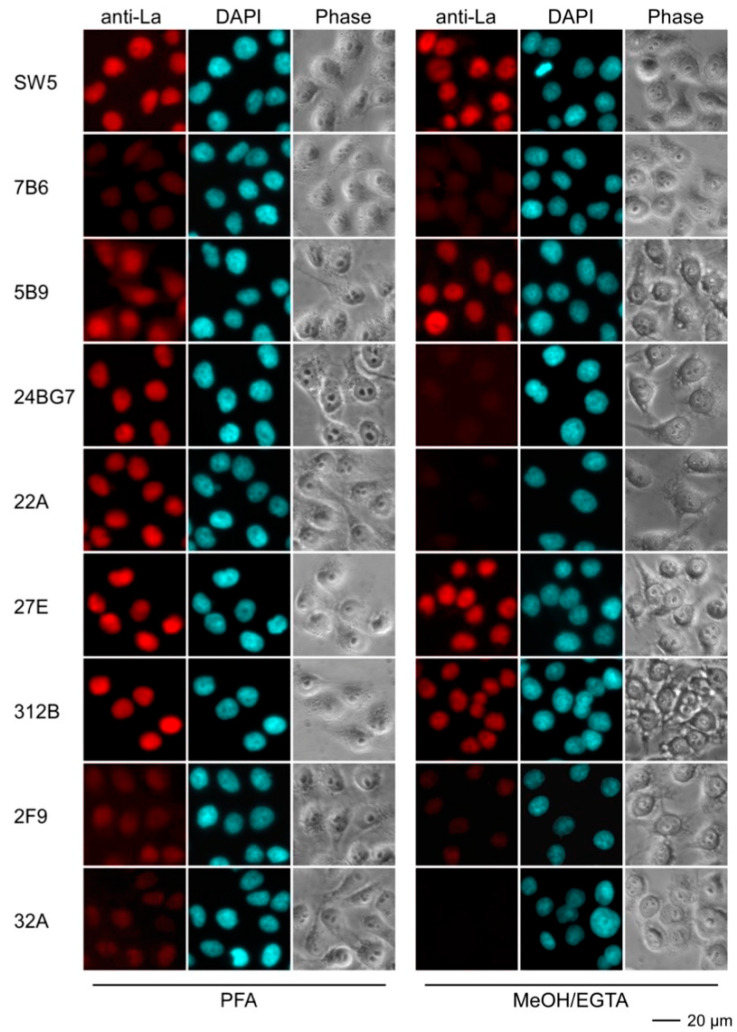
Epifluorescence microscopy of human HeLa cells stained with anti-La mabs. Prior to the staining, cells were fixed with either paraformaldehyde (PFA) or methanol (MeOH)/EGTA. For detection of the primary anti-La mabs, we used an anti-mouse-IgG ab conjugated with Alexa Fluor^®^568 as secondary ab. DNA was stained with DAPI. Phase, phase contrast image.

**Figure 9 ijms-22-01198-f009:**
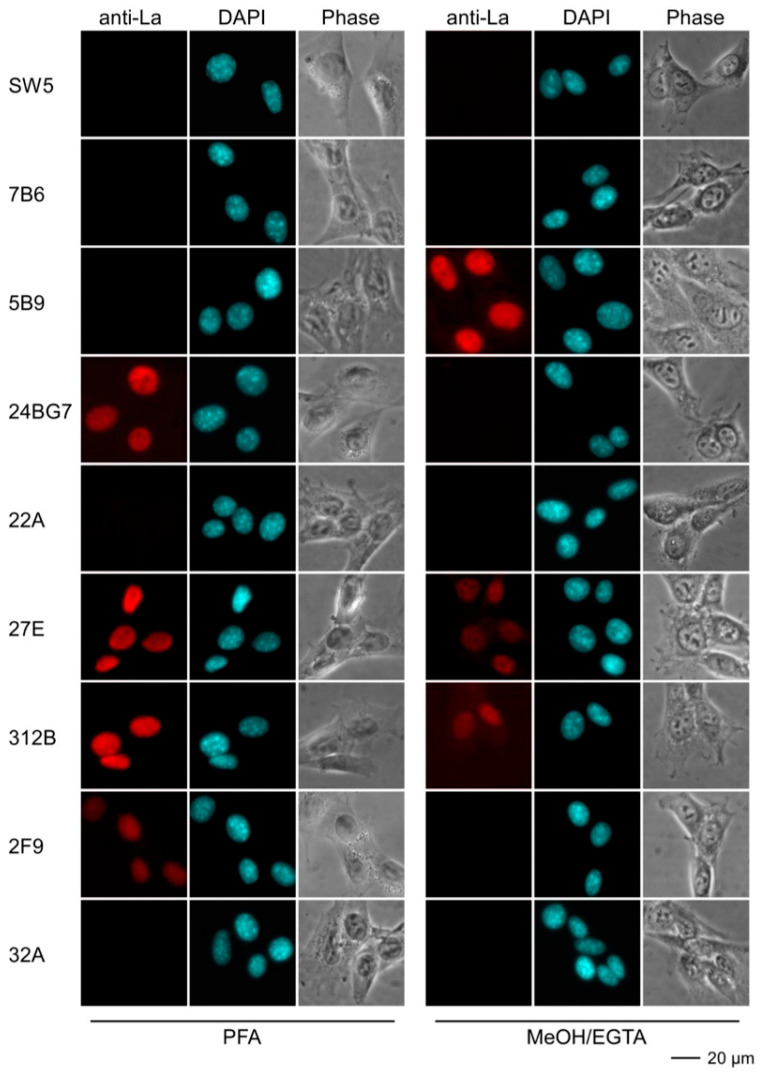
Epifluorescence microscopy of mouse 3T3 cells stained with anti-La mabs. Prior to the staining, cells were fixed with either paraformaldehyde (PFA) or methanol (MeOH)/EGTA. For detection of the primary anti-La mabs, we used an anti-mouse-IgG conjugated with Alexa Fluor^®^568 as secondary ab. DNA was stained with DAPI. Phase, phase contrast image.

**Figure 10 ijms-22-01198-f010:**
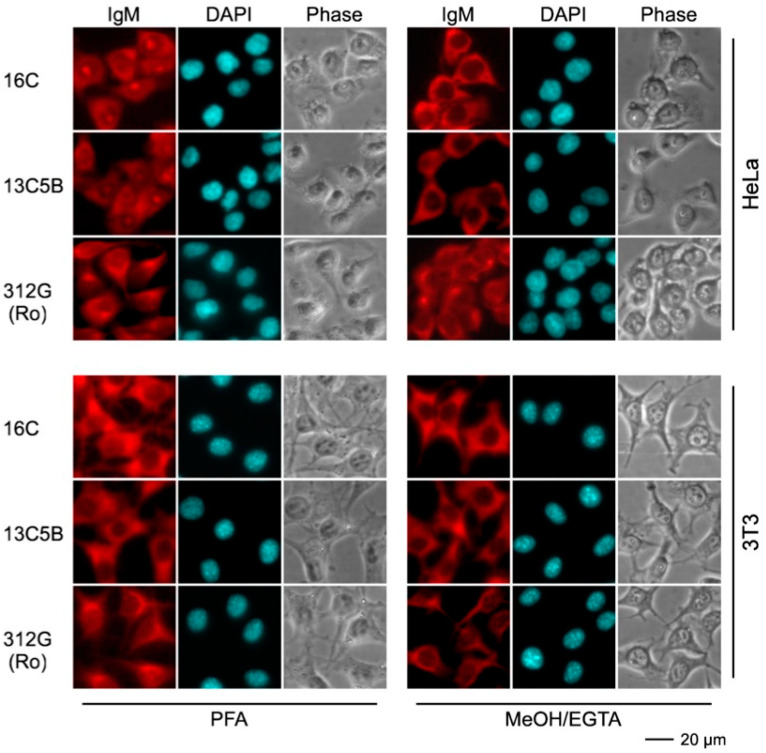
Epifluorescence microscopy of human HeLa cells and mouse 3T3 stained with anti-La IgM mabs. Prior to the staining, cells were fixed with either paraformaldehyde (PFA) or methanol (MeOH)/EGTA. For detection of the primary anti-La mabs (16C, 13C5B) or the anti-Ro mab (312G), we used an anti-mouse-IgM antibody conjugated with Alexa Fluor^®^568 as secondary ab. DNA was stained with DAPI. Phase, phase contrast image.

**Figure 11 ijms-22-01198-f011:**
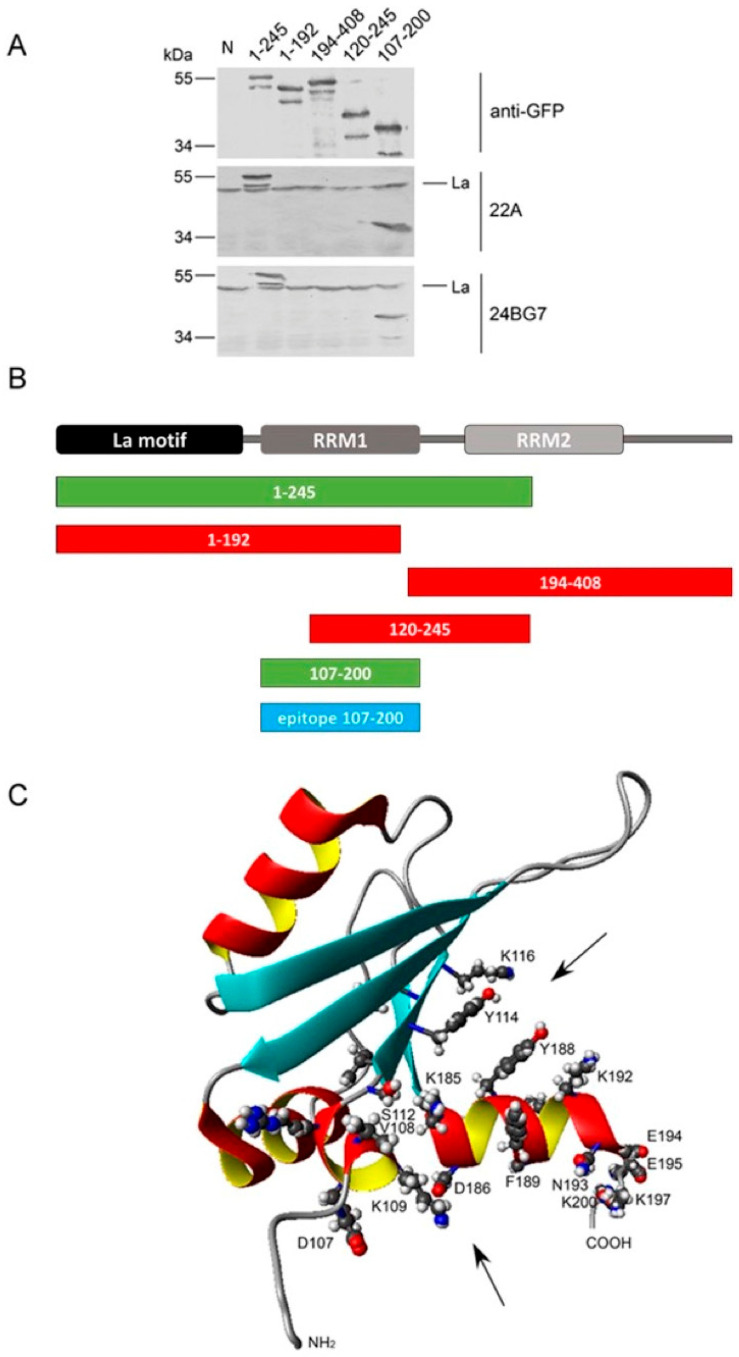
Epitope mapping of the Group 1 anti-La mabs (22A and 24BG7). HeLa cells were transfected with GFP-La fusion constructs encoding N- or C-terminally truncated La fusion proteins. Untransfected HeLa cells served as the negative control (N). (**A**) Total extracts were analyzed by SDS-PAGE and immunoblotting using antibodies against GFP or the anti-La mabs (22A or 24BG7); (**B**) Schematic summary of the selected GFP-La fragments. Immunoreactive GFP-La fusion proteins are highlighted in green. GFP-La fragments which failed to react with anti-La mabs are shown in red. The identified epitope region is highlighted in blue; (**C**) Predicted 3D structure of the RRM1 domain. Epitope mapping locates the epitope(s) recognized by the anti-La mabs (22A and 24BG7) to the RRM1 domain of the La protein. The folding of the RRM1 brings N- and C-terminal helices of the RRM1 in close vicinity, which may help to stabilize the 3D structure of the RRM1. The arrows point to potential conformational epitope regions.

**Figure 12 ijms-22-01198-f012:**
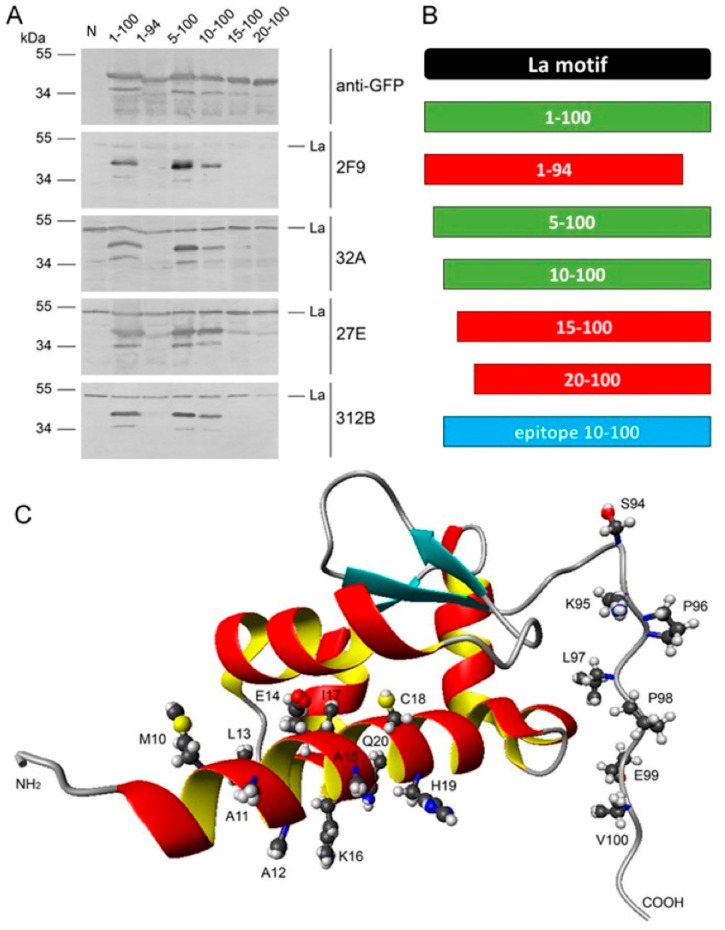
Epitope mapping of the Group 2 anti-La mabs (2F9, 32A, 27E and 312B). HeLa cells were transfected with green fluorescent protein (GFP)-La fusion constructs encoding N- or C-terminally truncated La fusion proteins. Untransfected HeLa cells served as the negative control (N). (**A**) Total extracts were analyzed by SDS-PAGE and immunoblotting using antibodies against GFP or the anti-La mabs (2F9, 32A, 27E und 312B); (**B**) Schematic summary of the selected GFP-La fragments. Immunoreactive GFP-La fusion proteins are highlighted in green. GFP-La fragments which failed to react with anti-La mabs are shown in red. The identified epitope region is highlighted in blue; (**C**) Predicted three-dimensional (3D) structure of the La motif. Epitope mapping locates the epitope(s) recognized by the anti-La mabs (2F9, 32A, 27E und 312B) to the La motif of the La protein. The N-terminal helix may contribute to the stabilization of the 3D structure of the La motif. Interestingly, the random coiled portion at the C-terminus represents the epitope recognized by the anti-La mab 5B9.

**Figure 13 ijms-22-01198-f013:**
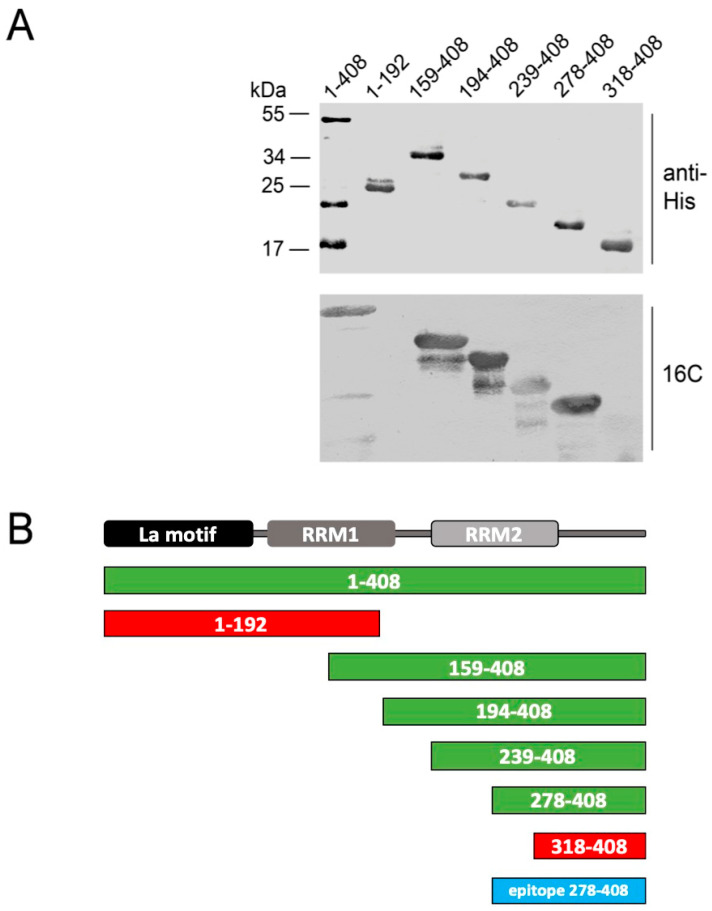
Epitope mapping of the IgM type anti-La mab (16C). Of all IgM type anti-La mabs, only epitope mapping data were obtained for the anti-La mab (16C). As it failed to react with HeLa cell extracts but reacted with full length La protein recombinantly expressed in *E. coli* (aa 1–408), N- and C-terminally truncated La deletion mutants were prepared, expressed in *E. coli,* and isolated via nickel-NTA affinity chromatography. (**A**) After SDS-PAGE and immunoblotting, the La deletion mutants were detected using either anti-His antibodies or the 16C anti-La mab. For detection of any immune reactivity of the anti-La mab (16C), we had to apply 10-fold the amount of protein as compared with the anti-His antibody; (**B**) Schematic summary of the selected La fragments. Immunoreactive La fragments are highlighted in green. La fragments which failed to react with the anti-La mab (16C) are shown in red. The identified epitope region is highlighted in blue.

**Figure 14 ijms-22-01198-f014:**
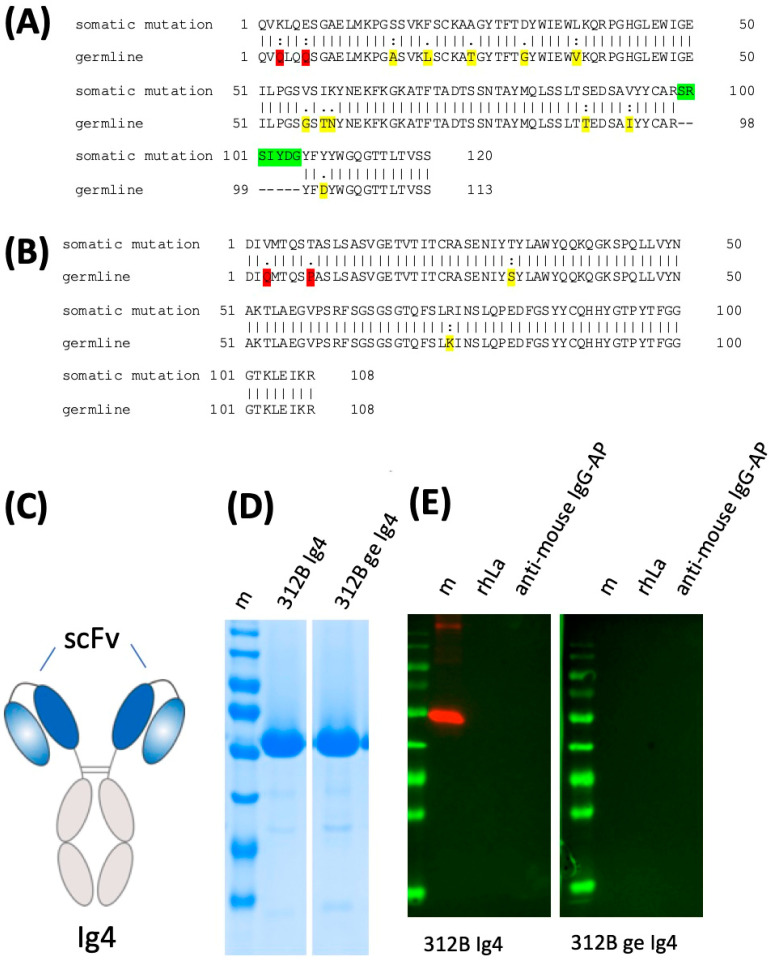
Lack of anti-La reactivity of the 312B germline sequence. (**A**) Alignment of the heavy chain sequence of 312B with the predicted germline sequence. Highlighted in yellow, the respective germline aa. Highlighted in red, mutations most likely caused during PCR due to the used degenerated forward primer. Highlighted in green, aa encoded by inserted n-nucleotides. Please note, we replaced the gap (dashed line) between the variable domain and the D element with this aa sequence also in the germline sequence; (**B**) Alignment of the light chain sequence of 312B with the predicted germline sequence. Highlighted in yellow, the respective germline aa. Highlighted in red, mutations most likely caused during PCR due to the used degenerated forward primer; (**C**) On the basis of the VDJ and VJ sequences of the mature 312B and the germline 312B, variable heavy and light chain domains scFvs were constructed and fused on an Ig4 heavy chain backbone; (**D**) Purified recombinant abs based on either the mature 312B sequence (312B Ig4) or the germline 312B sequence (312B ge Ig4) were separated by SDS-PAGE and stained with Coomassie brilliant blue; (**E**) SDS PAGE and immunoblotting analysis of recombinant human La (rhLa) with the ab based on the mature 312B sequence (312B Ig4) or the germline 312B sequence (312B ge Ig4). Marker proteins, m. Negative control, anti-mouse Ig conjugated with alkaline phosphatase (AP).

**Table 1 ijms-22-01198-t001:** Summary of identified VJ and VDJ elements. Estimated DNA sequences of the variable domains were compared with the NCBI data library using IgBlast sequence analysis tool. The name of the best fitting V, D, and J element (E) of the light (l) or heavy (h) chain is listed which was used for calculation of the mutation rate. %ID, percentage of identity. The estimated Ig isotype is also included.

mab	Isotype	V_l_		J_l_		V_h_		D_h_		J_h_	
		E	%ID	E	%ID	E	%ID	E	%ID	E	%ID
**SW5**	IgG2b	21–12	98%	JK1	86%	VH7183.a7.10	94%	DSP2.7 DSP2.5	91%	JH3	98%
**7B6**	IgG1	8–21	98%	JK1	100%	VHJ606.a6.127	88%	DSP2.8 DSP2.7 DSP2.5	100%	JH3	100%
**5B9**	IgG2a	8–21	98%	JK5	100%	J558.33	91%	DSP2.2	100%	JH3	100%
**24BG7**	IgG1	kk4	99%	JK2	100%	J558.22	94%	DST4.3 DST4.2	100%	JH3	100%
**22A**	IgG1	kk4	99%	JK5	94%	J558.80.186	89%	DSP2.9	89%	JH2	100%
**27E**	IgG2b	aa4	97%	JK1	100%	J558.22	93%	DSP2.3	100%	JH2	100%
**312B**	IgG1	12–44	98%	JK2	100%	J558.6.96	94%	DSP2.9	100%	JH2	96%
**2F9**	IgG1	21–5	99%	JK4	100%	3609.21.174	96%	DST4.3 DST4.2	100%	JH2	100%
**32A**	IgG1	aa4	99%	JK1	100%	J558.84.190	93%	DSP2.x	100%	JH2	100%
**16C**	IgM	bt20	99%	JK2	100%	J558.16.106	96%	DFL16.3	100%	JH4	100%
**13C5B**	IgM	aa4	98%	JK1	100%	J558.47	97%	DSP2.11 DSP2.10	100%	JH4	98%
**312G**	IgM	kk4	98%	JK5	100%	VHQ52.a19.61	98%	DFL16.3	100%	JH4	100%

**Table 2 ijms-22-01198-t002:** Analysis of mutation rate calculated for the respective light chain. The sequences of the V_l_ domains listed in Table 1 were compared and the number of mutations were calculated which occurred at the nucleotide (N) or amino acid level (A). Additional or missing nucleotides upstream of the J_I_ element are also given (n-nucleotides). The primer dependent first 24 nucleotides were not included in the analysis.

mab	Number of Mutations														n-Nucleotides
	FWR1		CDR1		FWR2		CDR2		FWR3		CDR3		FWR4		
	N	A	N	A	N	A	N	A	N	A	N	A	N	A	Upstream J_l_
**SW5**	1	1			2	2	1	1			1	1	4	3	+3
**7B6**			2	2					1	1					+2
**5B9**			1	1			1	1							+1
**24BG7**			1	1					1	1	1	1			−1
**22A**			2	1					1				2	2	+1
**27E**	2														
**312B**			1	1					1	1	1				+1
**2F9**									1						+1
**32A**	1	1							1	1					
**16C**															+2
**13C5B**															
**312G**															−1

**Table 3 ijms-22-01198-t003:** Analysis of mutation rate calculated for the respective heavy chain. The sequences of the V_h_ domains listed in Table 1. were compared and the number of mutations calculated which occurred at the nucleotide (N) or aa level (A). Additional or missing nucleotides upstream or downstream of the D_h_ element are also given (n-nucleotides). The primer dependent first 23 nucleotides were not included in the analysis.

mab	Number of Mutations												n-Nucleotides	
	FWR1		CDR1		FWR2		CDR2		FWR3		CDR3			
	N	A	N	A	N	A	N	A	N	A	N	A	Upstream D_h_	Downstream D_h_
**SW5**	1	1					8	5	8	3	2	2	+2	+1
**7B6**	5	2	2	2	5		13	8	6	2			+4	
**5B9**	3	1	6	4	4	1	6	5	3	1			+6	+6
**24BG7**	9	6	3	2			3	3	3	2			+3	+3
**22A**	3	3	3	2	2		12	7	8	4	1		+12	−3
**27E**	8	6	4	3			3	3	5	2			+5	+6
**312B**	3	3	2	1	1	1	3	3	4	2	2	1	+10	−4
**2F9**	1	1					1	1					−5	+13
**32A**	2		3	3			6	6	5	2				+6
**16C**	3	2					2	2	1	1			+7	+6
**13C5B**	1	1	3	2	1	1			2		1	1	+1	+5
**312G**									2	1			+4	+5

**Table 4 ijms-22-01198-t004:** Summary of reactivities of the mabs obtained by immunoblotting, coprecipitation, and epifluorescence microscopy. Semiquantitative evaluation of reactivities, (+++) strong, (++) medium, (+) low, ((+)) weak, (-) negative. C, cytoplasmic. (*) not working.

mab	Reactivity Recombinant Proteins Immunoblotting				Reactivity m3T3 hLa Immunoblotting		Immuno-Precipitation m3T3 hLa		Immuno-Fluorescence PFA		Immuno-Fluorescence MeOH	
	rhLa	rmLa	HeLa	3T3	hLa	mLa	hLa	mLa	HeLa	3T3	HeLa	3T3
**SW5**	+++	++	+++	-	+++	-	++	-	+++	-	++	-
**7B6**	+++	++	+++	-	++	-	-	-	(+)	-	(+)	-
**5B9**	+++	+++	+++	+++	+++	+++	-	-	++	-	+++	+++
**24BG7**	+++	+	+++	+	++	+	+	-	++	++	-	-
**22A**	+++	-	+++	-	+	-	++	-	++	-	-	-
**27E**	+++	+	+++	+++	++	++	+	+	+++	+++	+++	+
**312B**	+++	+++	+++	+++	+	++	+++	+++	+++	+++	++	+
**2F9**	+++	+++	++	++	+	+	-	-	+	+	(+)	-
**32A**	+++	-	++	-	(+)	-	-	-	(+)	-	-	-
**16C**	+	*	*	*	*	*	*	*	C	C	C	C
**13C5B**	+	*	*	*	*	*	*	*	C	C	C	C
**312G**	-	*	*	*	*	*	*	*	C	C	C	C

## Data Availability

The data presented in this study are available on request from the corresponding author.

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
