# Peer review of "T Cell Mediated Conversion of a Non-Anti-La Reactive B Cell to an Autoreactive Anti-La B Cell by Somatic Hypermutation†"

_ijms, 2021, doi:10.3390/ijms22031198_

Round 1
Reviewer 1 Report
Although the research question may not catch most of the readers' interest and the clinical application of the data is very limited, the methodology of the experiments is sound, the results and conclusions are solid. Minor comments are:
1) The whole manuscript is rather long and difficult to follow. The authors are encouraged to substantially shorten it and eliminate repetitive descriptions.
2) It seems that some sections may have lost. Section 3.4 follows that of 2.3. Please check if there is information missing.
3) While the method of T-cell adoptive transfer is often referenced (number 51), this reviewer would like to review again exactly how the authors harvested and isolated the T cells from the non-transgenic mice, and transferred to the transgenic mice. Information should include (but not exclusive) to what kind of T cells are isolated and transferred, the number of T cells being transferred each time and the frequency of transfer.
Author Response
Rev1.
Reviewer 1: Although the research question may not catch most of the readers' interest and the clinical application of the data is very limited, the methodology of the experiments is sound, the results and conclusions are solid. Minor comments are:
Reviewer 1: 1) The whole manuscript is rather long and difficult to follow. The authors are encouraged to substantially shorten it and eliminate repetitive descriptions.
Response 1) We are grateful for the overall positive evaluation of our work. As mentioned in the cover letter in parallel to the original submission of the ms, the data presented in the ms were collected over the past 15 years. One can imagine that we have collected plenty of additional data during this time. The current ms is therefore the first but very important ms of a series of following up ms. Over time the whole story is way more complex as we thought when we started with our work. However, we didn’t want to start with publishing until we understand the full picture. Already while writing the submitted version of the ms we prepared several different versions of it. In order to reduce the complexity we had to shorten it several times. This may also be a reason why our data appear of limited clinical relevance in the reviewer’s view although we think that understanding of the mechanism how an autoimmune response is triggered should also be of great interest for the understanding of the disease and the development of novel future treatment strategies. Anyway, we also thought of further shortening of the ms e.g. by moving half of the IF data to a supplemental data section and/or putting the sequence comparisons into a supplemental section. However, we finally decided not to do such a further shortening because it will be much easier to practically use these data having the images/sequence data directly site by site. Bearing in mind that Rev 2 seems to well accept the length and format of the ms we therefore prefer not to further condense the paper and we hope that our concerns about shortening can be accepted.
Reviewer1: 2) It seems that some sections may have lost. Section 3.4 follows that of 2.3. Please check if there is information missing.
Response 2) We apologize for this mistake and are grateful that the reviewer did see this error. The section 2.4 and the following sections 2.5 and 2.6 were wrongly labelled as 3.4 to 3.6. We have corrected this mistake in the revised ms.
Reviewer1: 3) While the method of T-cell adoptive transfer is often referenced (number 51), this reviewer would like to review again exactly how the authors harvested and isolated the T cells from the non-transgenic mice, and transferred to the transgenic mice. Information should include (but not exclusive) to what kind of T cells are isolated and transferred, the number of T cells being transferred each time and the frequency of transfer.
Response 3) In general T cell isolation and adoptive transfer was performed as described in reference 51. In order to fulfill the request of the reviewer, however, we added the following more detailed description of the T cell isolation and transfer (see lanes 644 to 666 of the revised ms highlighted in red):
In summary, donor mice were immunized with 100 micrograms 6xhis-human recombinant La protein in Complete Freund’s Adjuvant (Difco, KS, U.S.A.) delivered in a divided dose subcutaneously in one hind footpad and the base of tail. Seven days later, T cells were enriched from draining inguinal and popliteal lymph nodes by nylon wool chromatography from four animals. Residual B cells and other antigen presenting cells were depleted using a 1:1 mixture of Dynabeads (Dynal, Great Neck, NY, U.S.A.) coated with anti-B220 (Dynal 114.01) and anti-rat IgG Dynabeads coated with anti-mouse I-A/I-E antibody (Clone 10.2.16). Such isolated cells should represent a mixture of both CD4+ and CD8+ T cells but were not further characterized. The recipient La transgenic mouse received 5 million of the isolated T cells intraperitoneally. As control, T cells were isolated from similarly treated animals with the only difference that the immunization mixture lacked recombinant human La protein. Isolated T cells were also adoptively transferred in another La transgenic mouse. Hybridomas were prepared from both animals. All the herein described anti-La hybridomas were obtained from the La transgenic recipient mouse which received the T cells isolated from non-transgenic mice after immunization with recombinant human La protein. In contrast, we could not isolate a single anti-La hybridoma from the control mouse indicating that the anti-La reactive B cells were attracted to the spleen by the adoptively transferred anti-La reactive T cells present in the T cell preparation isolated from the non-La transgenic mice after immunization with human recombinant La protein.
Reviewer 2 Report
The paper herein is nice and overall well-written. I suggest to include following references:
Massicotte H, Harley JB, Bell DA. Characterization of human-human hybridoma monoclonal anti-Ro(SS-A) autoantibodies derived from normal tonsil lymphoid cells. J Autoimmun. 1992 Dec;5(6):771-85.
Pan ZJ, Davis K, Maier S, Bachmann MP, Kim-Howard XR, Keech C, Gordon TP, McCluskey J, Farris AD. Neo-epitopes are required for immunogenicity of the La/SS-B nuclear antigen in the context of late apoptotic cells. Clin Exp Immunol. 2006 Feb;143(2):237-48.
Author Response
Reviewer 2: Comments and Suggestions for Authors
The paper herein is nice and overall well-written. I suggest to include following references:
Pan ZJ, Davis K, Maier S, Bachmann MP, Kim-Howard XR, Keech C, Gordon TP, McCluskey J, Farris AD. Neo-epitopes are required for immunogenicity of the La/SS-B nuclear antigen in the context of late apoptotic cells. Clin Exp Immunol. 2006 Feb;143(2):237-48.
Massicotte H, Harley JB, Bell DA. Characterization of human-human hybridoma monoclonal anti-Ro(SS-A) autoantibodies derived from normal tonsil lymphoid cells. J Autoimmun. 1992 Dec;5(6):771-85.
Response Reviewer 2: We are very grateful for these kind comments of the reviewer. As requested we have added both references. They became references 55 and 56. As a consequence of adding of these references the reference numbering had to be modified. The original references 55 to 61 became references 57 to 63. We also had to add few sentences to justify the addition of these two references (lanes 515 to 521).